# How Chemotherapy Affects the Tumor Immune Microenvironment: A Narrative Review

**DOI:** 10.3390/biomedicines10081822

**Published:** 2022-07-28

**Authors:** Marco Carlo Merlano, Nerina Denaro, Danilo Galizia, Fiorella Ruatta, Marcella Occelli, Silvia Minei, Andrea Abbona, Matteo Paccagnella, Michele Ghidini, Ornella Garrone

**Affiliations:** 1Scientific Direction, Candiolo Cancer Institute, FPO-IRCCS Candiolo, 10060 Torino, Italy; 2Department of Medical Oncology, Fondazione IRCCS Ca’ Granda Ospedale Maggiore Policlinico, 20122 Milano, Italy; nerinadenaro@gmail.com (N.D.); fiorella.ruatta@gmail.com (F.R.); michele.ghidini@policlinico.mi.it (M.G.); ornella.garrone@policlinico.mi.it (O.G.); 3Multidisciplinary Oncology Outpatient Clinic, Candiolo Cancer Institute FPO-IRCCS, 10060 Candiolo, Italy; danilo.galizia@ircc.it; 4Department of Medical Oncology, S. Croce e Carle Teaching Hospital, 12100 Cuneo, Italy; occelli.m@ospedale.cuneo.it; 5Post-Graduate School of Specialization Medical Oncology, University of Bari “A.Moro”, 70120 Bari, Italy; silvia.minei3@gmail.com; 6Division of Medical Oncology, A.O.U. Consorziale Policlinico di Bari, 70120 Bari, Italy; 7Translational Oncology ARCO Foundation, 12100 Cuneo, Italy; abbona.andrea@gmail.com (A.A.); matteo.babeuf@gmail.com (M.P.)

**Keywords:** chemotherapy, immune cells, tumor microenvironment

## Abstract

Chemotherapy is much more effective in immunocompetent mice than in immunodeficient ones, and it is now acknowledged that an efficient immune system is necessary to optimize chemotherapy activity and efficacy. Furthermore, chemotherapy itself may reinvigorate immune response in different ways: by targeting cancer cells through the induction of cell stress, the release of damage signals and the induction of immunogenic cell death, by targeting immune cells, inhibiting immune suppressive cells and/or activating immune effector cells; and by targeting the host physiology through changes in the balance of gut microbiome. All these effects acting on immune and non-immune components interfere with the tumor microenvironment, leading to the different activity and efficacy of treatments. This article describes the correlation between chemotherapy and the immune changes induced in the tumor microenvironment. Our ultimate aim is to pave the way for the identification of the best drugs or combinations, the doses, the schedules and the right sequences to use when chemotherapy is combined with immunotherapy.

## 1. Introduction

Evidence that an efficient immune system is important to optimize the effect of chemotherapy arises from experimental studies in mouse models published many years ago [1].

Patients affected by acquired immunodeficiency syndrome who developed cancers usually controlled by chemotherapy were refractory to treatment, particularly before the availability of the antiretroviral drugs [2].

Indeed, human immunosuppressive virus (HIV) is linked to CD4 expressed on T-helper cells, inducing cell apoptosis and leading to a defective immune response [3].

Therefore, it is now widely accepted that a competent immune system is required to optimize chemotherapy effects.

In addition, high immune infiltrate is associated with a favorable prognosis in many tumors, giving further evidence to the role of the immune system on the patient’s outcome [4,5].

Many studies have investigated the effects of conventional drugs on the components of the tumor microenvironment (TME). It has been suggested that conventional drugs may reactivate immunosurveillance due to (but not limited to) the induction of immunogenic cell death, and these properties may represent a good starting point for the design of rational immunotherapy–chemotherapy combinations [6,7,8].

Today there is an increasing belief that chemotherapy can favor the antitumor immune response and may represent a scientifically based partner for immunotherapy [9,10].

In particular, the immune effects of chemotherapy seem more evident when drugs are delivered below the maximum tolerated dose or in a metronomic way [11,12].

Chemotherapy exerts its immune effects by targeting the tumor, immune cells, host physiology and other aspects of the TME.

More recently, it has been realized that targeted therapeutic drugs developed to block specific pathways may induce off-target effects able to interfere with the TME, and these will be briefly considered in the present review.

## 2. Chemotherapy

### 2.1. Targeting Cancer Cells

Chemotherapy induces cell stress, leading to damaged signal pathway activation. Under these conditions, apoptotic cancer cells translocate chaperon proteins related to the endoplasmic reticulum (ER), such as calreticulin, heat shock protein (HSP)-70 and HSP-90, to the cell membrane surface, while cytoplasmic proteins such as the chromatin-associated high-mobility group box 1 (HMGB-1) and purine metabolites, such as adenosine triphosphate (ATP), are released in the extracellular space [13]. ATP is a “find me signal” that promotes the recruitment of immature dendritic cells (DC). The ER chaperons, not expressed on the membrane surface under normal conditions, are “eat me signals” which facilitate the phagocytosis of cancer cells by immature DCs, a key point for the identification and elaboration of foreign antigens. HMGB-1 promotes DC maturation after binding to TLR-4.

All the previous described molecules are markers of immunogenic cell death (ICD), a type of programmed cell death favoring the recognition of cancer by the immune system. The same mechanism is used by the immune system to recognize infected cells. 

The large amount of DNA and RNA fragments (known as ‘damage-associated molecular profiles’—DAMPs) in the extracellular space arising from dying cancer cells bind to toll-like receptor (TLR)-3 and TLR-9 which activate the production of type 1 interferons and other pro-inflammatory cytokines such as IL-6, IL-12 and TNF-α, which induce an inflammatory microenvironment and the activation of the innate immune response [14,15].

Beyond TLR, DAMPs bind other families of membrane receptors (pattern recognition receptors—PRR) including: TLR, NOD-like receptors (NLR), RIG-I-like receptors (RLR) and C-type lectin receptors (CLR). Among these, TLRs are the most important in onco-immunology due to their greater ability to bind DAMPs.

In addition, other cytoplasmatic receptors act as DNA sensors, such as stimulators of interferon genes (STING) [16].

DAMPs also include extracellular signals, for instance, matrix fragments generated by proteolytic enzymes such as hyaluronic acid or biglycan released from dying cells [16].

Following activation, PRRs trigger the production of many pro-inflammatory cytokines which, in turn, activate the innate immune response.

All these events ultimately promote the presentation of tumor antigens to naive CD8+ and CD4+ T lymphocytes, triggering the adaptive response (Figure 1).

DAMP-initiated inflammation, independent of viral or bacterial infection, is called “sterile inflammation” [17].

Apoptotic cell death may be immunogenic, but not all apoptotic cell deaths are immunogenic.

Indeed, immunogenic apoptosis requires to be preceded by the previously described cell stress responses [12].

This explains why physiologically dying cells, such as epithelial cells, do not induce an immunogenic response.

In addition to apoptosis, other mechanisms of programmed cell death include necroptosis and pyroptosis [18].

Pyroptosis is a programmed cell death induced by infections. Pyroptosis is Caspase 1-dependent, but is not dependent on the activation of Caspase 3, 6 or 8, which are critical drivers of apoptotic death [19].

Necroptosis is a form of caspase-independent, highly inflammatory, programmed cell death. Whether necroptosis is more immunogenic compared to apoptosis is unknown; however, it is a common opinion that necroptosis might be more immunogenic [20].

Despite this, cancer cells killed by chemotherapy mostly die from apoptosis [21].

On the contrary, the combination of chemotherapy and radiotherapy might induce cell killing mainly through necroptosis. For instance, we have shown in the p53 wild-type F9 murine teratocarcinoma cell line treated with concurrent cisplatin and radiation that among dying cells necrosis is much more common than apoptosis [22]. Later studies have confirmed that radiation alone may induce necroptosis [23,24], supporting the hypothesis of the high immunogenic role of radiation therapy.

Drugs structurally similar, such as platinum salts or taxanes, have different capability to induce ICD (Table 1) [25].

### 2.2. Targeting Immune Cells

Each cancer drug may have an effect on immune cells in addition to the ability of inducing ICD. Many reports in recent years have faced this issue [7,8,9,10,11,12]. However, many data arise from experimental, in vivo and/or in vitro models.

In the following pages, we report a summary of these effects, focusing on data available from humans or from experimental models.

There are two main ways to restore immune response against tumor cells by chemotherapy: (a) the inhibition of the immune suppressive cells or (b) the activation of immune effector cells.

#### 2.2.1. Inhibition of Immune-Suppressive Cells

Experimental data show both in vitro and in vivo, but even in humans, that a lot of drugs can act on immunosuppressive cells. Among them, the optimal targets are T regulatory cells (Treg), myeloid-derived suppressor cells (MDSCs), tumor-associated macrophages (TAMs) M2 and cancer-associated fibroblasts (CAFs).

Tregs are among the principal actors in the tumor immune suppressive microenvironment. Indeed, one of the effects associated with the recovery of the antitumor response is the increase in the ratio of effector T cells to regulatory T cells as observed both in humans and in experimental models [12,26].

Cyclophosphamide administered at a metronomic dose selectively depletes Treg cells without affecting CD 8+ Teff cells, as demonstrated by Ghiringhelli et al. in a series of ten end-stage patients affected by different types of solid tumors [27].

Dimeloe et al. studied human T cells and observed that the cyclophosphamide-extruding transporter ABCB1 is less expressed in Treg than in Teff cells, supporting the observed selective effect of metronomic cyclophosphamide [28].

MDSCs, a heterogeneous family of immature immune cells, include two main subsets: granulocytic (gMDSC) and monocytic (mMDSC) cells. The former are predominant in humans (about 75% of all MDSCs), while mMDSCs are less represented and more immunosuppressive [29].

However, unlike physiological conditions, the two subsets are differently expressed among human solid tumors. For instance, mMDSCs are dominant in melanoma [29], while gMDSCs are dominant in renal cell carcinoma [30].

Chemotherapy affects MDSCs in an ambiguous way. Indeed, chemotherapy may increase circulating gMDSCs but not mMDSCs in breast cancer patients treated with neo-adjuvant chemotherapy [31], while in patients with glioblastoma, metronomic capecitabine significantly reduces circulating MDSCs without specifying the effect on the two subsets [32].

Gemcitabine selectively depletes gMDSCs in patients with pancreatic cancer [33].

Targeting TAM-M2, the pro-tumor macrophage phenotype, is an important method of chemotherapy to restore the immune response. Chemotherapy may change TAM polarization toward the M1 (the antitumor phenotype), exploiting TAM plasticity.

Chemotherapy is also able to recruit monocytes or macrophages to the tumor and deplete TAM-M2 in the TME [34].

Cyclophosphamide, cisplatin, carboplatin and paclitaxel are among the cytotoxic drugs able to induce TAM reprogramming from the M2 to the M1 phenotype in experimental models and in patients [34].

Paclitaxel demonstrates the ability to reprogram M2-polarized macrophages to the M1-like phenotype in vitro and in vivo through binding TLR4, which leads TAM to promote pro-inflammatory signals and cytokines which activate other immune cells, such as dendritic cells. This effect has been observed in experimental models and is supported by changes in gene expression in tumor samples from patients with ovarian cancer before and after treatment [35].

Many anticancer drugs mimic a battlefield, inducing damaged/dying cells and recruiting monocytes to the tumor bed as a nonhealing wound must be repaired [34,36]. The recruitment of immature macrophages in the tumor may reduce tumor growth, as shown by Laoui et al. based on clinical and experimental data [37], although in the TME macrophages tend to switch to the M2 phenotype. However, the final effect is not clear and deserves more studies.

Trabectedin, in addition to its cytotoxic effect on cancer cells through DNA binding [38], shows selective cytotoxicity against macrophages: a dramatic decrease in TAM density has been observed in patients with sarcomas after neoadjuvant therapy [39]. In the mouse model, trabectedin reduced chemokine ligand (CCL) 2 levels, which are crucial for monocyte recruitment.

#### 2.2.2. Activation of Immune Effector Cells

Gemcitabine restores the proliferative capacity of T effector cells, resulting in an increase in effector T (Teff) cells/Tregs ratio [33].

Dendritic cells have a key role in activating (or re-activating) the immune response and represent the link between innate and adaptive immunity.

As shown above, DCs are activated both indirectly, by chemotherapy through ICD, and directly. An in vitro study has demonstrated that some agents, such as anti-microtubule drugs, antimetabolites, antitumor antibiotics and paclitaxel administered at low metronomic doses, improve DC maturation and function. These effects have been observed in vitro in human DCs [40].

In particular, paclitaxel may up-regulate CD 80, CD 86 and CD 40, triggering the second activating signal in T cells through the binding of CD 80 and CD 86 to CD 28.

Finally, binding CD 40 on DC with CD 40L on T helper is necessary to authorize DC to prime naïve CD8+ T cells [41].

Moreover, low-dose paclitaxel in an in vivo model enhances the maturation of DC precursors by the activation of TLR-4 and ultimately favors the efficient priming of CD8+ T cells [42].

In conclusion, the effects of chemotherapy on immune cells are determined by the selected drug and scheduling. Moreover, each drug is able to induce peculiar effects that are in part or completely different drug by drug.

Table 2 summarizes the effects induced by some chemotherapy drugs on immune cells.

Albeit not strictly related to the topic of this review, it is important to underline for clarity that all the positive effects of chemotherapy on the TME can also be counteracted by the TME itself. For instance, in experimental models, ATP released by apoptotic cells (a typical effect of chemotherapy as reported above) may be converted into adenosine by CD 39 and CD 73 [43] which potently suppresses T cell immunity. Clinical studies are in progress to evaluate the role of targeting CD 73, CD 39 and the adenosine A2 receptor (A2AR) in cancer immunotherapy [44].

### 2.3. Effect of Chemotherapy on the Expression of PD-L1

Cytotoxic drugs favor the release of danger signals in the tumor microenvironment and induce an inflammatory milieu. Therefore, they may up-regulate PD-L1 expression [45].

In turn, PD-L1 contributes to chemoresistance through extracellular signal-regulated kinase (ERK) hyperactivation [46]. Indeed, ERK is a well-known actor of multidrug resistance.

On the contrary, PD-L1 up-regulation sensitizes cancer cells to immunotherapy. Interestingly, PD-L1 overexpression also involves chemo-resistant cells [47] and favors response to immunotherapy even in subsets of patients expressing resistance to chemotherapy.

We have seen that chemotherapy may increase anti-tumor immune response through different complementary mechanisms.

Most of these mechanisms reactivate the anti-tumor immune response through the inhibition of immunosuppression. Immunosuppression is a physiological event to limit the damage induced by the prolonged inflammatory status of healthy tissues exploited by the tumor to avoid the control of the immune system. However, after the reactivation of the immune–inflammatory response, the immune system in turn up-regulates the immunosuppressive mechanisms to repress the inflammatory response, following the same physiological pathway. The up-regulation of PD-L1 is a key point of this process which paves the way to the combination of chemotherapy and immunotherapy either concurrently or in sequence.

It has been observed in many tumors, including non-small cell lung cancer [48] and head and neck cancer [49,50], that immunotherapy favors subsequent response to chemotherapy after treatment failure.

The biological basis in support of this observation may be based on the cross-talk PD-L1-ERK; the former is up-regulated by chemotherapy, the latter by PD-L1 itself [50]. After chemotherapy exposure, the tumor may over-express PD-L1 which, in turn, may up-regulate ERK, favoring multidrug resistance. Treatment with PD-(L)1 axis inhibitors may down-regulate PD-L1 expression [51,52] and might reduce ERK expression, restoring chemosensitivity [46].

### 2.4. Targeting the Host Physiology

It is known that chemotherapy causes gastrointestinal toxicity, including nausea, vomiting and diarrhea. These effects can modify the body mass index and the composition of gut microbiota.

Gut microbiota is made up of commensal bacteria, viruses, fungi and archaea. These microorganisms live in the human digestive system and affect human health.

Its composition is modulated by several non-modifiable factors, such as age, sex and host genomics, and modifiable factors such as diet, body mass index, medication and directly by chemotherapy [53].

Changes in gut microbiota lead to dysbiosis that may influence response to chemotherapy [54] and also may induce resistance to immune checkpoint inhibitors, as observed in patients with lung and renal cancer receiving broad-spectrum antibiotics [55]. Intriguingly, data show that antibiotic therapy administered within one month of starting immunotherapy is the worst predictor of response compared to antibiotic therapy given more than one month before immunotherapy [56] and even during immunotherapy [57].

Currently, researchers have not yet demonstrated the ideal composition of a “favorable” microbiota [58].

In patients with melanoma, the composition of gut microbiome is associated with a better response to anti-PD-1 immunotherapy [59]. Based on these data, probiotics (live bacteria) have been advertised as health-promoting agents and widely sold as over-the-counter medications.

In this case, the customer’s choice is driven by marketing and costs rather than scientific data. This may be risky.

Indeed, some reports have shown that the consumption of commercial probiotics reduces response to immunotherapy in melanoma patients [60,61].

Although there is still no ideal composition of a favorable microbiota or knowledge on how to modulate it, Lee et al. have recently published a series of recommendations on how to maintain or restore good gut microbiota in patients receiving immunotherapy (Table 3) [62].

## 3. Targeted Therapeutic Drugs

Although this review focuses on the effects of chemotherapy, it should be highlighted that many targeted agents used in clinical practice are able to modulate TME, as well as the classical anticancer drugs and their effects deserve to be known at least in general terms.

For instance, there is growing evidence that cyclin-dependent kinase (CDK) 4/6 inhibitors mediate immunomodulatory effects [63].

BRAF-, KRAS- and PI3K/AKT-driven tumors support an immunosuppressive microenvironment. Accordingly, specific inhibitory agents, (vemurafenib, dabrafenib, trametinib and others) mediate different immunostimulatory effects [64].

Human epidermal growth factor receptor-2 (HER-2), epidermal growth factor receptor (EGFR) and vascular endothelial growth factor-A (VEGFA) are among the earliest molecular targets that have attracted the interest of the scientific community. Many tyrosine kinase inhibitors (TKI) and monoclonal antibodies (Mab) have been developed and licensed during the past three decades.

Table 4 summarizes some immunologic effects of selected targeted therapeutic drugs in common clinical use.

## 4. Conclusions

Chemotherapy increases tumor immunogenicity through several complementary mechanisms.

All these mechanisms aim to improve the antitumor immune response by inhibiting immunosuppression or boosting immune effector cells.

Altogether, the immunomodulatory effects of chemotherapy could be harnessed to design biologically based combined treatments.

It is already possible to find clinical studies on different solid tumors showing significant benefits from the combination of chemotherapy and immunotherapy compared to single treatment. These studies are generally designed by adding standard chemotherapy regimens to immunotherapy.

However, to maximize the effect, chemotherapy could also be adapted to exploit its immunological effects. This may require a modification of the dose or scheduling. In other words, the combination should not be the simple sum of two effective treatments administered together in a conventional way because greater benefits could be obtained by reconsidering all biological effects.

A limitation of this approach is that many data on the immunologic effects of chemotherapy arise from experimental models, and only a few of them have been verified in humans.

However, the possibility of making the most of the treatments already available represents a great opportunity that justifies the effort to extend and validate our knowledge developed in “in vivo” and “in vitro” models in humans.

In this way, we will be able to identify the best drug or combination, the best dosage, schedule, and the right sequence to be used with immunotherapy.

## Figures and Tables

**Figure 1 biomedicines-10-01822-f001:**
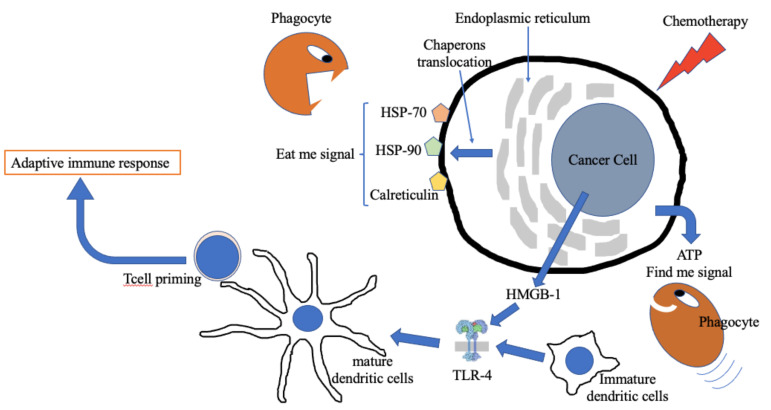
Immunogenic cell death. Under chemotherapy attack, cancer cells initiate a series of modifications favoring the activation of the immune system. ATP is released in the microenvironment by injured cells, representing a “find me signal” able to attract phagocytes. Many chaperons expressed in the endoplasmic reticulum translocate to the cell membrane representing an “eat me signal” that can be recognized by phagocytes. HMGB1 released in the extracellular fluid under cell stress conditions favors dendritic cell maturation via TLR-4 activation. HSP = heat shock protein; ATP = adenosine triphosphate; TLR = toll-like receptor; DC = dendritic cells; HMGB-1 = high-mobility group box-1.

**Table 1 biomedicines-10-01822-t001:** The induction of main ICD-associated DAMPs.

Drug	CRT Exposure	ATP Release	HMGB1 Release
Cisplatin	-	+	+
Carboplatin	±	+	±
Oxaliplatin	+	+	?
Docetaxel	+	-	+
Paclitaxel	+	+	+

CRT = calreticulin; ATP = adenosine triphosphate; HMGB1 = chromatin-associated high-mobility group box 1; ? = unknown; Data from Fumet J-D et al. [25].

**Table 2 biomedicines-10-01822-t002:** Effects of selected chemotherapy drugs on immune cells.

Immune Cells	Drugs	Effect	Issues	Model	Ref.
T-reg	mCTX	Depletion	Dose and scheduling dependent	Human	[27] [28]
MDSC	Cape Gem	Depletion	MDSC includes gMDSC and mMDSC Selectivity of active drugs is unclear	Human	[32] [33]
TAM-M2	CTX CDDP CBDCA PCTXL	Reprogramming toward M1 phenotype		Human	[34] [35]
	Trabectidin	Depletion			[39]
DC	PCTXL	Maturation	Experimental data	In vivo and in vitro	[42]
CD 8+ T cells	Gem	Proliferation	-	Human	[33]

Legend: T-reg = T regulatory cell; MDSC = myeloid-derived suppressor cells; TAM-M2 = tumor-associated macrophage—M2; CAF = cancer-associated fibroblast; DC = dendritic cell; mCTX = metronomic cyclophosphamide; Cape = capecitabine; Gem = gemcitabine; CTX = cyclophosphamide; CDDP = cisplatin; CBDCA = carboplatin; PCTXL = paclitaxel; nPCTXL = nab-paclitaxel; gMDSC = granulocytic MDSC; mMDSC = myelocitic MDSC.

**Table 3 biomedicines-10-01822-t003:** Dietary and general main recommendations for patients receiving immunotherapy.

Dietary	
Diversifying the diet	Consumption of a great variety of different foods
Having a high fiber intake	At least 30 g/day
Consuming many different plant species	30 different species/week recommended
Advising patients	Against consumption of self-prescribed commercially available probiotic supplements
**General**	
Broad-spectrum antibiotics	Treatment, especially one month before starting immunotherapy, should be avoided unless strictly necessary
If antibiotics needed	A microbiology consultation should be required to avoid broad-spectrum antibiotics
If broad-spectrum antibiotics given within one month from the planned immune treatment	Consider temporarily delaying the start of immunotherapy

Data from Lee K.A. 2021[58].

**Table 4 biomedicines-10-01822-t004:** Immune effects of selected targeted drugs licensed for clinical use.

Target	Agent	Effect	Ref.
CDK 4/6	Abemaciclib	Improved antigen presentation; pro-inflammatory cytokines release; depleted Treg	[65]
	Palbociclib	Improved antigen presentation; pro-inflammatory cytokines release; PD-L1 up-regulation; activation of Teff cells; depleted Treg	[66]
	Ribociclib	Improved antigen presentation	[66]
BRAF	Dabrafenib	Improved antigen presentation; enhanced Teff functions	[67]
	Vemurafenib	Improved antigen presentation; enhanced Teff functions	
MEK	Trametinib	Improved antigen presentation; ICD;	[68]
	Cobimetinib	Activation of Teff cells	[69]
PI3K	Alpelisib	Improved antigen presentation; PD-L1 down-regulation	[70]
EGFR	Cetuximab	Improved antigen presentation; ADCC; ICD; TAM-M2 polarization	[71]
	Gefitinib	Improved antigen presentation; PD-L1 down-regulation; DC activation	[72]
	Erlotinib	Improved antigen presentation	
	Afatinib	Improved antigen presentation	
HER-2	Trastuzumab	Improved antigen presentation; DC activation; ADCC; TAM-M2 polarization	[64]
	Pertuzumab	Improved antigen presentation; ADCC	
VEGF	Bevacizumab	Teff expansion; DC activation	[73]
	Apatinib	PD-L1 down-regulation	[74]
	Sunitinib	Treg depletion; MDSC depletion	[75]

Treg = T regulatory cells; Teff = T effector cells; ICD = immunogenic cell death; ADCC = antibody-dependent cell cytotoxicity; TAM = tumor-associated macrophages; DC = dendritic cell.

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
