# Peer review of "How Chemotherapy Affects the Tumor Immune Microenvironment: A Narrative Review"

_biomedicines, 2022, doi:10.3390/biomedicines10081822_

Round 1

Reviewer 1 Report

In this manuscript, a basic overview of chemotherapy impact on immune cells present in tumor microenvironment is given. The main aim of the manuscript is to highlight how chemotherapy has the potential to trigger and reinvigorate the immune system, which would ultimately support the process of tumor eradication.

Despite the author interest in the field, the manuscript requires a considerable improvement before its publication.

Abstract

Line 19 – “However” is more used in opposition to a previous claim. Here I would rather use “Furthermore”.

Introduction

Overall, the introduction should be improved in the way it is written, at the moment each reference is followed, and in its content (there are more information in the abstract than in this section).

Line 35. “In human” can be removed since the terms “patient” already highlight the organism.

Line 40. “in good shape” doesn’t sound scientific. I would suggest “(…) a fit/competent immune system”.

Core of manuscript

Line 55. “Chemotherapy induces cell stress and stressed cells, in turn, activate damage signals pathways.” The sentence can be rephrased as “Chemotherapy induces cell stress, leading to damage signals pathways activation”.

Line 56. “stress” is overused, “Under these conditions” is better.

Line 61. “the so called” doesn’t sound scientific. I suggest “known as”.

Line 67. TLR is already mentioned before, I would suggest starting the paragraph with “Beyond TLR,”.

Line 71 to 78. These sentences are not bound together.

Line 88. Necessary?

Line 97. Not clear.

Line 127. In vivo and in vitro should be in italic. Also in other sections of the manuscript (e.g. Line 134).

Section 5, about PD-L1, should go before section 4., about physiology.

Final remarks

Figure name and abbreviations should be placed under the relative image (same for tables). Legend of figure 1 should also be followed by a description. Figure one should have more chromatic difference. The grid of the tables should be visible, making it easier to be analyzed.

Many concepts within the same paragraphs appear unbound to each other and presented as a list of concepts (e.g. Line 71 to 78 and Line 243 to 245).

The language doesn’t always sound scientific (e.g. in good shape, so called).

The author should also add a section explaining when the effects mediated by chemotherapy on immune response do not work. For instance, it is known that CD39 and CD73 mediate the conversion of ATP derived from apoptotic cells (typically tumor cells under chemotherapy) into adenosine, ultimately affecting the efficacy of cytotoxic T cells. Overall, tumor microenvironment is also able to protect tumor cells from the effect of chemotherapy.

Author Response

We thank the Reviewer for the comments which improved the quality of the manuscript.

Abstract:

"however" was deleted and "furthermore" used

Introduction:

We have enlarged some aspects of the introduction also following the suggestions of the other Reviewers. We have also deleted the term "in human" and "in good shape" (rephrased as suggested in "a competent immune system")

Core of manuscript:

"Chemotherapy induces cell stress and stressed cells..."  has been rephrased as suggested: "chemotherapy induces cell stress, leading to..."

The overused term "stress" has been changed in "Under these conditions..."

The phrase "the so called" has been changed in "Known as"

We have started the paragraph using "Beyond TLR,"

We agree that the sentences reported at the lines 71 - 78 (and in particular lines 73 - 78) in the original manuscript are not bound together. This aspect in our opinion is due to a bad sequence of the talk. For this reason we have moved the lines 73 - 78 in the edited manuscript before "The large amount of DNA and RNA fragments...". We hope that this change has clarified the narration.

The DAMPs initiated inflammation... is called "sterile inflammation". The Reviewer query is "Necessary?". Of course it is not for people familiar with this topic. However we hope that this paper, if accepted for publication, will be read also by colleagues that are not familiar with the argument and therefore we believe usuful to maintain this sentence.

"immunogenic cell death applies to apoptotic..." we have modified the sentence to improve clarity

In vitro and in vivo has been changed in in vitro and in vivo

Section on PD-L1 has been moved bifore section on physiology, as requested

Final remarks:

Figure name and abbreviations have been placed under the image and tables; figure 1 is now followed by a description and the figure has been redesigned with more chromatic differences. The grid of the tables are now visible

We have already detailed the changes made about lines 71 - 78; about lines 243 - 245, (the choice is driven by marketing and costs...) we have rephrased the sentence (...has been advertised ass health promoting agents and widely sold as over-the counter medications. In this case the customer's choice...)

The language has been corrected to sound more scientific

We had a section to explain that under particular circumstances immune effects mediated by chemotherapy do not work (Albeit not strictly related to the topic of this review, it is important to underline...)

Reviewer 2 Report

The manuscript "How chemotherapy affects the tumor immune microenvironment" is intriguing, well written and clearly explains how chemotherapeutic drugs modulate immune system cells in the tumor microenvironment. I have some minor suggestions for improvement of the article:

 In Table 1, include and discuss other classes of drugs also.

Check the manuscript for minor grammatical mistakes e.g. comma, hyphen, capitalization

For clarity rephrase lines 99, 106, 211 (depend), 227 (viruses), 269.

Avoid new paragraphs if continuing the same idea.

The author should include and discuss the recently discovered and used anti-cancer drugs like targeted therapeutic drugs and antibody-based therapeutic drugs apart from conventional chemotherapeutic drugs. Some of them can be found in the following articles.

  Liu H, Sun S, Wang G, Lu M, Zhang X, Wei X, Gao X, Huang C, Li Z, Zheng J and Zhang Q (2021) Tyrosine Kinase Inhibitor Cabozantinib Inhibits Murine Renal Cancer by Activating Innate and Adaptive Immunity. Front. Oncol. 11:663517. doi: 10.3389/fonc.2021.663517

 Gaggianesi M, Di Franco S, Pantina VD, Porcelli G, D'Accardo C, Verona F, Veschi V, Colarossi L, Faldetta N, Pistone G, Bongiorno MR, Todaro M and Stassi G (2021) Messing Up the Cancer Stem Cell Chemoresistance Mechanisms Supported by Tumor Microenvironment. Front. Oncol. 11:702642. doi: 10.3389/fonc.2021.702642

 Di Ianni N, Maffezzini M, Eoli M and Pellegatta S (2021) Revisiting the Immunological Aspects of Temozolomide Considering the Genetic Landscape and the Immune Microenvironment Composition of Glioblastoma. Front. Oncol. 11:747690. doi: 10.3389/fonc.2021.747690

 Liu, Q., Liao, Q. & Zhao, Y. Chemotherapy and tumor microenvironment of pancreatic cancer. Cancer Cell Int 17, 68 (2017). https://doi.org/10.1186/s12935-017-0437-3

Author Response

We thank the Reviewer for the comments.

"In table 1 include other classes of drugs too":

We have included a new section (Targeted therapeutic drugs) as suggested by the Reviewer in a following comment ("The Authors should include and discuss the recently discovered and used...). We added a table (Table 4) devoted specifically to this topic, rather then add these drugs in Table 1. In our opinion this choice may look more easy for the Readers.

"For clarity rephrase lines 99 (has been made also following Reviwer#1) 106 (rephrased) 211 ("depend") (changed in "is determined by") 227 (virus) (corrected in "viruses") 269 (rephrased all the sentence "...after the reactivation of the immune-inflammatory response...)

Reviewer 3 Report

Although the intention of the authors to structure this very broad topic, some of the grouping is misleading: discussing the role of CAFs under the topic “targeting immune cells” is confusing. In addition, they cannot be clearly identified as immunosuppressive cells (3.1. subheading). A less ambiguous structure would be very helpful.

Although the overview of the different studies is very helpful in this paragraph 3.1 and 3.2, a high-level summary of the different studies is missing.

The authors state that they focus on clinical studies. Deviating therefrom they mention preclinical studies and sometimes it is hard to understand if the data they discuss are from or preclinical or clinical studies. A clear distinction would increase the value of the review.

Line 115: necrosis might be a typo instead necroptosis makes more sense

Author Response

We thank the Reviewer for comments and suggestions.

"...discussing the role of CAF under the topic "targeting immune cells" is confusing..."

We agree with the Reviewer and we have changed in large part the section devoted to CAFs. We hope that in the present form, the narrative is less ambiguous 

"Although an overview of the different studies...paragraph 3.1 and 3.2"

We have provided more informations regarding the studies mentioned in the manuscript. We have also clarified if they have been conducted in experimental models, in clinic or both, to answer the last question of the Reviewer ("Deviating therefrom, they mention preclinical studies...")

Round 2

Reviewer 1 Report

The author improved the manuscript and answered my questions and concerns.

Minor change

Line 73. "All the previous described proteins (...)". ATP is not a protein... Please exchange "proteins" with "molecules".

Author Response

line 73: we have changed "proteins " with "molecules".

We thank the Reviewer

Reviewer 3 Report

the authors addressed all comments despite the section about CAFs. The scientific landscape here is much more complex than they anticipate Either they expand the topic of the review or they skip the paragraph about CAFs completely

Author Response

We have skipped the paragraph about CAF.

We agree with the Reviewer about the complexity of this topic.

We just underline that almost all the topics faced in this review are much more complex than described. For instance, the allocation of Tumour Associated  Macrophages in two sub-populations (M1 and M2), albeit largely adopted by most Authors, is an oversimplification since M2 TAM includes at least 4 sub-populations (Pan Y, 2020; Wu K, 2020). In our opinion, simplification is necessary when a review faces many aspects of tumour immunology. However we agree that CAF represents a highly complex topic, and therefore we have followed the suggestion of the Reviewer, and we have deleted the paragraph.